# A Review of Recent Pharmacological Advances in the Management of Diabetes-Associated Peripheral Neuropathy

**DOI:** 10.3390/ph16060801

**Published:** 2023-05-29

**Authors:** Osman Syed, Predrag Jancic, Nebojsa Nick Knezevic

**Affiliations:** 1Advocate Illinois Masonic Medical Center, Department of Anesthesiology, Chicago, IL 60657, USA; osman.syed@midwestern.edu (O.S.); prjancic@gmail.com (P.J.); 2Chicago College of Osteopathic Medicine, Midwestern University, Downers Grove, IL 60515, USA; 3Department of Anesthesiology, University of Illinois, Chicago, IL 60612, USA; 4Department of Surgery, University of Illinois, Chicago, IL 60612, USA

**Keywords:** diabetic peripheral neuropathy, diabetes, neuropathic pain, GLP-1

## Abstract

Diabetic peripheral neuropathy is a common complication of longstanding diabetes mellitus. These neuropathies can present in various forms, and with the increasing prevalence of diabetes mellitus, a subsequent increase in peripheral neuropathy cases has been noted. Peripheral neuropathy has a significant societal and economic burden, with patients requiring concomitant medication and often experiencing a decline in their quality of life. There is currently a wide variety of pharmacological interventions, including serotonin norepinephrine reuptake inhibitors, gapentanoids, sodium channel blockers, and tricyclic antidepressants. These medications will be discussed, as well as their respective efficacies. Recent advances in the treatment of diabetes mellitus with incretin system-modulating drugs, specifically glucagon-like peptide-1 agonists, have been promising, and their potential implication in the treatment of peripheral diabetic neuropathy is discussed in this review.

## 1. Introduction

Diabetic peripheral neuropathy (DPN) is defined as a disorder primarily associated with diabetes mellitus without any evident correlation to other causes of peripheral nerve damage [1]. The plethora of symptoms attributed to DPN is explained by the extensive amount of diabetic neuropathy (DN) forms, including polyneuropathy, mononeuropathy, radiculoplexopathy, autonomic neuropathy, and others [2]. However, an absence of symptoms does not exclude DN, as asymptomatic presentation is common [3]. Of the vast variety of presenting DN forms, distal sensorimotor polyneuropathy (DSMP) overshadows other entities, accounting for 75–90% of DPN cases [2,4].

The aim of this review paper is to highlight the current available knowledge of DPN pain and to shine a light on the direction in which clinical trials are headed with pharmacological treatment.

## 2. Clinical Characteristics

Clinical findings on DSMP can be roughly categorized into two groups, positive and negative. The hallmark of positive findings is pain, followed by allodynia (regular stimuli such as touch and pressure cause pain) and hyperalgesia (increased pain intensity to a stimulus that usually causes less pain) [5]. Of the patients with DSMP, up to 25% may experience pain as their main symptom [3,4,6,7]. 

Negative symptoms include decreased sensation to touch, temperature, and pain, which can result in ataxia and other motor disabilities [8,9].

Pain in diabetic patients can be categorized into non-neuropathic and neuropathic. Due to a complex array of factors included in the pathophysiology of diabetes, non-neuropathic pain can arise from different vascular, osteochondral, and neural structures (peripheral vascular disease, spinal stenosis, radiculopathy, etc.) [2]. Successfully distinguishing between these two types of pain is essential for diagnosing and treating neuropathic pain. Moreover, it is suggested that pain of neuropathic origin is greater in intensity and less likely to respond to medication compared to non-neuropathic pain [10].

Neuropathic pain presents symmetrically and distally, and is usually associated with burning, prickling, and shooting sensations. This clinical presentation is often a sign of small nerve fiber involvement [3,4,6,8]. Damage to larger nerves may give rise to predominantly negative symptoms (numb feet, loss of sensation), which puts patients at risk of further complications, such as feet deformities and ulcerations [11]. Large fiber damage can also present with ataxic gait and frequent falls [12,13].

## 3. Epidemiology

In the setting of urbanization and inadequate dietary trends, the prevalence of diabetes mellitus and its complications is on a steep incline [14]. Recent findings suggest that 8.5% of the world population has a diagnosis of diabetes, with that number expected to exponentially increase in the coming years [15]. A 2018 report shows that 23.4 million people in the US have diagnosed diabetes. Moreover, 81.6 million US citizens (33.9%) have prediabetes [16].

A concerning finding shows that 17.7% of US adolescents aged 12–19 have prediabetes, whereas 28.5% of those with diabetes are undiagnosed [16]. A staggering 22.9% of people with diabetes are undiagnosed and untreated in the US [16]. Taking these numbers into account, it is easily deduced that the incidence of diabetic complications is on the rise, and more focus should be placed on the prevention of such complications, rather than treatment [4].

Due to the varying degree of clinical presentation, lack of awareness of the disease progression, and an unclear standard in diagnostic protocols, the prevention and treatment of DPN is often challenging [3,17,18,19]. With these challenges in mind, making a concrete statement regarding the prevalence of diabetic complications is often difficult. This difficulty is in part due to the multifactorial pathogenesis of such complications. Studies have found that, besides glycemia, key players in neuropathy formation include hypertension, smoking, obesity, and elevated triglyceride levels [20].

The estimated prevalence of DSMP varies in the literature, but studies suggest that up to 15% of newly diagnosed cases of diabetes have concomitant DSMP, with that number rising up to 50% during the 10-year progression of the disease [4,21]. Certain papers estimate that 28% of diabetic patients in a primary care setting have DSMP, with this number being approximately 20% for the total diabetic population [7,22].

Among patients with type 1 diabetes, the prevalence of DSMP is somewhat misleading. One study showed that 8.2% of diagnosed youth had DSMP, whereas the prevalence in adolescents with type 2 diabetes paralleled that in the adult population (25.7%) [23]. However, other studies reported higher numbers for type 1 diabetes-associated DSMP, ranging from 23% [24] to 27% [25]. 

Concerning primarily type 1 diabetes, an association worth noting is between puberty and the onset of diabetic complications. Studies have suggested that the hormonal changes happening throughout puberty lead to a variety of pathophysiological processes that can result in a significant increase in incidence of diabetic complications [26,27,28].

## 4. Societal and Economic Burden

It is of no surprise that chronic pain leaves a trail through multiple facets of a person’s life, including mental health and well-being. A staggering statistic shows that 43% of patients with DSMP use concomitant medication (for anxiety, depression, sleep deprivation, etc.), illustrating the beginning of the humanitarian burden that this disease brings [29].

When scoring DSMP patients on a scale that represents the interference of pain in their everyday activities, there is a trend of higher scores, corresponding to an increased interference rate [30]. Moreover, scales testing for daily living disability, quality of sleep, and frequency of anxiety and depression among patients show the same trend [30,31].

Diabetic neuropathy and its complications contribute to an estimated 27% of total annual costs aimed at diabetes management [32]. Therefore, the economic burden of DSMP is two-fold. Not only is the cost of polypharmacy required for these patients considerable, but the productivity lost in a society due to these disabilities is substantial [31].

## 5. Pharmacological Management in Diabetic Peripheral Neuropathy

The management of DPN is a complicated subject with cultural and patient-specific nuances. For the purposes of this review article, the “standards of care” shall be drawn from the 2022 American Academy of Neurology’s (AAN) practice guidelines [33], as well as the 2023 American Diabetes Association (ADA) standards of care [34]. This review article will provide a brief synopsis of these medications as well as attempt to elucidate the usage of new medications in the treatment and prevention of DPN. Therefore, the two main categories of medications reviewed in this paper are symptomatic treatments, focusing on the treatment of pain in DPN (SNRIs, gabapentinoids, sodium channel blockers, and TCAs), and pathogenesis-based treatments (GLP-1 agonists). A summary of the included systematic reviews, meta-analysis, and studies is shown in Table 1. 

### 5.1. Symptomatic Therapy

#### 5.1.1. Serotonin Norepinephrine Reuptake Inhibitors (SNRIs)

Patients often respond well to SNRIs, and they are considered a first-line treatment by the AAN and the ADA. Both societies specifically mention venlafaxine, desvenlafaxine, and duloxetine as SNRI agents with varying levels of efficacy.

SNRIs work by inhibiting the serotonin transporter (SERT) and norepinephrine transporter (NET) proteins, which are responsible for removing these neurotransmitters from the synapse after they have been released. Additionally, SNRIs are believed to indirectly modulate other neurotransmitter systems, such as dopamine and acetylcholine, which may also contribute to their therapeutic effects [53]. 

In a Cochrane systematic review, a group of five trials had results favoring the usage of duloxetine compared to a placebo. Patients were pooled from the trials across multiple doses: 20 mg, 40 mg, 60 mg, or 120 mg per day. The review found that the relative rate of greater than 50% improvement in pain was 1.53 (95% CI). The paper further specified that the 20 mg-treated patients did not have a statistically significant change, but there was a paucity of data as only one trial had a 20 mg arm [35].

The statistical review carried out by the AAN in the writing of their guidelines found that duloxetine is moderately effective with moderate confidence, while desvenlafaxine was also found to be moderately effective with lower confidence [33].

A 2021 meta-analysis comparing the efficacy of duloxetine to gabapentin found no significant difference between the medications in terms of management of DPN pain. Moreover, the study suggested that the choice between these two groups is mostly based on cost-effectiveness (SNRIs being more expensive than gabapentionids), patients’ reports of side effects, and personal preference [36].

A 2023 systematic review showed similar results, further confirming the safety and efficacy of duloxetine in DPN pain treatment. The review also suggested that, considering the shorter list of side effects compared to pregabalin, it may serve as a preferential treatment. In addition, the review found that a dosage of duloxetine of 120 mg, compared to the regular 60 mg, can be given to patients to ensure pain relief without change in the frequency of side effect reports, further confirming its safety [37]. A recent systematic review found comparable results, proposing duloxetine as a preferential drug in comparison to gabapentin, due to the same levels of efficacy but better duloxetine tolerance [38].

It is worth noting that a 2017 meta-analysis comparing local capsaicin use to oral pain relief medication showed promising results. The study analyzed 25 randomized controlled trials, suggesting that a capsaicin 8% patch provides pain relief similar to that of oral duloxetine, while also being an adequate substitute for pregabalin and gabapentin. Moreover, the study suggests that the patches may have a better side-effect profile compared to oral medication, in part due to the method of administration [39].

#### 5.1.2. Gabapentinoids

Gabapentinoids are a class of drugs that include gabapentin and pregabalin. The mechanism of action of gabapentinoids is thought to involve modulation of the activity of voltage-gated calcium channels. Specifically, gabapentinoids bind to the alpha2-delta subunit of these channels, which reduces calcium influx into neurons and thereby decreases the release of excitatory neurotransmitters such as glutamate and substance P [54]. This effect may account for their anticonvulsant properties as well as their ability to reduce neuropathic pain. Additionally, gabapentinoids have been shown to increase the synthesis and release of GABA, the primary inhibitory neurotransmitter in the central nervous system. This effect may also contribute to their anticonvulsant properties and may explain why they are effective in treating anxiety disorders [54]. 

The efficacy of pregabalin, which is FDA-approved for the treatment of painful diabetic neuropathy, has been shown to be high for both the management of pain of common comorbidities that arise due to DPN, such as sleep interference [55]. Gabapentin has also been shown to have a pain-reducing effect, with one multicenter RCT showing a mean relief of 39% after 8 weeks [56].

Mirogabalin (DS-5565) is a new gabapentinoid recently brought onto the market in Japan. The drug has the same mechanism of action as other gabapentinoid medications, but has increased potency at the alpha2-delta subunit, as compared to pregabalin [57]. The following clinical trials indicate efficacy in the treatment of DPN. 

A randomized double-blind trial specifically looking at patients with DPN showed that doses of mirogabalin between 15 and 30 mg/day led to significant reductions in pain when compared to a placebo at the 5-week mark [40]. The trial also had a single arm with patients randomized to 300 mg of pregabalin, and found no significant improvement in pain for patients. This trial can be found in Table 2, with other recently completed trials researching DPN pain. 

Another randomized, double-blind trial was conducted with 834 patients, all of whom had DPN. Patients were randomized to either a placebo group or a group who received a mirogabalin dose between 15 and 30 mg/day. In all groups, the average daily pain score did decrease over the course of the 14-week trial, but the final results only showed a statistically significant decrease for the group randomized to the 30 mg/day dose [41].

So far, all clinical trials involving mirogabalin have been performed in East Asian countries, and the medication is approved for usage in Japan. At this time, there is no indication if mirogabalin will undergo clinical trial for DPN elsewhere.

#### 5.1.3. Sodium Channel Blockers

Sodium channel blockers are a class of drugs that target the voltage-gated sodium channels found in neurons and cardiac cells. These channels are responsible for generating and propagating action potentials, which are essential for the transmission of signals in the nervous and cardiovascular systems. Sodium channel blockers work by binding to the channel pore and preventing the influx of sodium ions, which results in the inhibition of action potential generation and propagation. Sodium channel blockers that are currently used in the treatment of pain are drugs that are also used as anesthetics, class 1 antiarrhythmics, anti-convulsants (such as oxcarbazepine), and tricyclic antidepressants [58].

Sodium channel inhibition can be selective or non-selective, depending on the specific drug and the type of sodium channel being targeted. For example, some drugs, such as lidocaine, selectively block channels that are in an activated or inactivated state.

On the other hand, drugs such as tetrodotoxin (TTX) selectively inhibit specific isoforms of sodium channels. This discovery led to the categorization of nine known isoforms of sodium channels into two groups, TTX-sensitive and TTX-resistant. Isoforms Na_V_1.1, Na_V_1.2, Na_V_1.3, Na_V_1.4, Na_V_1.6, and Na_V_1.7 are TTX-sensitive, with their predominant location being on the central and peripheral neurons, with the exception of Na_V_1.4 being mostly found in skeletal muscle. The remaining three isoforms are TTX-resistant and expressed in cardiac muscle (Na_V_1.5) and the dorsal root ganglion neurons (Na_V_1.8 and Na_V_1.9) [59]. This range in sensitivity amongst the sodium channels is due to a difference in the amino acid sequence that makes the binding site for TTX, leaving some of them resistant to this sodium channel inhibitor [60].

While in some clinical applications, broad-spectrum blockade is favorable, only certain sodium channels have been implicated in nociceptive pathways and, therefore, are preferable to selectively inhibit to avoid systemic side effects [58].

A Na_V_1.7 isoform of the sodium channel has been the target of extensive research in previous years due to the fact that a loss of function mutation of the gene that encodes this isoform results in total body insensitivity to pain. This discovery lead researchers to believe that a selective inhibition of this channel might prove beneficial in pain treatment in various pathological states [61,62].

Multiple studies have suggested that a group of molecules called arylsulfonamides could play a significant role in finding a specific inhibitor of the Na_V_1.7 isoform [63,64]. Moreover, certain sodium channel blockers such as lidocaine have been shown to enhance the inhibitory effect arylsulfonamides have on Na_V_1.7 channels. [65] Due to the very similar structure of these channels and their abundance in different tissues throughout the body, the search for a specific inhibitor is proving to be difficult, and more extensive research is needed in the field.

Both the AAN and ADA have sodium channel blockers as listed therapy options for the treatment of DPN, with the AAN specifically stating they have a moderate confidence in the data cited in their review [33].

A 2020 systematic review analyzing 43 randomized controlled trials showed that a lidocaine 700 mg medicated plaster was equivalently efficient in peripheral neuropathic pain management compared to pregabalin, yet had a better adverse event profile [42].

#### 5.1.4. Tricyclic Antidepressants

Tricyclic antidepressants (TCAs) are a class of drugs that were among the first antidepressants to be developed and widely used. The mechanism of action of TCAs is complex, but they primarily act by inhibiting the reuptake of the monoamine neurotransmitters, including norepinephrine, serotonin, and dopamine, back into presynaptic neurons. This results in an increased concentration of these neurotransmitters in the synaptic cleft, which enhances neurotransmission and mood stabilization. Additionally, TCAs are known to block various receptors, including histamine, alpha-adrenergic, and muscarinic receptors, which may contribute to their clinical effects and side effects [43].

TCAs are considered second-line to SNRIs when it comes to neuropathic pain [66]. TCAs can be successfully used as monotherapy for neuropathic pain, but their adverse event profile can be more burdensome to patients compared to SNRIs [67]. Moreover, a dose greater than 75 mg daily is not recommended for patients over the age of 65 because of the dose-dependent anticholinergic side effects and an increased risk of falling [68].

Amongst a vast body of literature detailing this point, a newer 2020 systematic review of 18 placebo-controlled trials further confirmed that SNRIs are a preferential treatment for neuropathic pain, leaving TCAs as an alternative option [43].

A 2022 network meta-analysis looking at fibromyalgia treatment further corroborates these statements, showing that even though amitriptyline was associated with improvement in quality of life, fatigue, and pain, to some degree, duloxetine 120 mg (SNRI) was found to be more beneficial [44].

In a recent meta-analysis, 16 out of 18 placebo-controlled trial comparisons were positive for TCA use in chronic neuropathic pain, with the combined number needed to treat being 3.6, further confirming the use of TCAs in these patients [45].

The 2022 multicenter, crossover OPTION-DM trial studied the efficacy of different combinations of drugs used in the treatment of DPN. Over the course of 50 weeks, all patients enrolled underwent therapy with different treatment pathways, testing both monotherapies and combination therapy if patients did not have adequate pain relief. The study showed that supplementing amitriptyline with pregabalin (amongst other supplementation combinations) resulted in no statistically significant difference in diabetic neuropathy pain management outcome compared to other treatment combinations (including amitriptyline, pregabalin, and duloxetine). Furthermore, the study illustrated that current therapeutics alone are not adequate for a large number of patients, as monotherapy did not provide significant pain relief for about two thirds of trial participants. Although the study did give credibility to the usage of combination therapy, as it was generally more efficacious, the study also showed that a significant number of patients did not reach adequate levels of pain reduction with any combination, showing the need for new therapies [46].

### 5.2. Pathogenesis-Based Therapy

#### 5.2.1. GLP-1

The options available in the treatment of diabetes have recently expanded with glucagon-like peptide-1 (GLP-1) agonists, which operate via the modulation of the incretin hormonal system. This system normally operates via the enteroendocrine system in a periprandial fashion. As a bolus of food is ingested, GLP-1 is released from secretory granules by intestinal L cells both through a neural signaling pathway via gastrin-releasing peptide (GRP) and acetylcholine, and eventually through L-cell direct contact interaction with the bolus of food. GLP-1 itself then can act on the GLP-1 receptor (GLP-1R), which is found on pancreatic islets, as well as throughout the GI tract, the vagus nerve, hypothalamus, and brainstem. Action on the pancreas leads to a stimulation of insulin release, while simultaneously inhibiting glucagon release and slowing gastric emptying. GLP-1 is then degraded by dipeptidyl peptidase-4 (DPP-4), as well as other endopeptidases [69].

Therefore, drugs that act to modulate this system either operate by mimicking GLP-1 or delaying endogenous GLP-1 degradation via inhibition of DPP-4. Drugs recently brought to market include semaglutide, lixisenatide, dulaglutide, liraglutide, and exenatide.

Tirzepatide is a GLP-1 agonist that is a modified analog of gastric inhibitory peptide (GIP). It has been demonstrated that tirzepatide has stronger affinity for the GIP receptor than the GLP-1 receptor. This dual mechanism of action still acts to promote insulin secretion, but has different pharmacodynamic properties, which have been shown to be beneficial in terms of improving insulin sensitivity, as well as reducing obesity [70].

Furthermore, retatrutide, a GLP/GIP/glucagon triple agonist, with limited but promising data, is another medication being brought to market [71]. As with the current treatments for DPN, future trials testing the effects of these multi-receptor incretin mimetics on DPN are still needed.

If a provider feels that a patient is a good candidate for GLP-1 agonist treatment, then lab testing and patient history guide the selection of which specific agent is appropriate. Half-lives vary dramatically, with current market formulations lasting between 2.4 h and a week [72]. A review of different GLP-1 agonists found that longer-acting agents are more effective at improving glycemic control.

A meta-analysis of 34 RCTs showed that GLP-1 agonists of all formulations are very effective at lowering HbA1c, with treatments ranging between 0.55% and 1.21%; dulaglutide and liraglutide had the greatest reduction of 1.21% and 1.15% on average, respectively [47].

##### Possible Role of GLP-1 Agonists in the Treatment/Prevention of Ongoing Diabetic Neuropathy

GLP-1 agonists have recently come onto the market for the treatment of diabetes, and as previously mentioned, DPN is a frequent complication of diabetes. A 2022 meta-analysis that included 101,440 patients treated with either sodium–glucose co-transporter 2 (SGLT-2) inhibitors or GLP-1 agonists was studied to see if there were major adverse limb events in either group. After analysis was performed, the authors concluded that GLP-1 agonist use can be associated with significantly reduced risks of adverse limb events [48].

Another 2022 meta-analysis aimed to compare lower extremity amputation rates in patients treated with SGLT-2 inhibitors versus DPP-4 inhibitors and GLP-1 agonists. Based on their analysis of eight retrospective case–control designs, they found no advantage of either category in terms of limb loss rates [73]. In the following sections, the methods by which GLP-1 agonists may contribute to the treatment of DPN are outlined.

##### Microvascular Disease

In the pathogenesis of diabetes patients, high circulating sugars can lead to oxidative stress, as previously mentioned, but also to dysfunction of the endothelium in blood vessels. This process can occur anywhere, but would be of significant concern in the microvasculature that feeds nerve fibers. This process can lead to ischemia of the nerve, causing functional changes within the nerve, and eventually loss of the nerve fiber itself. Although peripheral neuropathy is the most common, autonomic and cranial nerve neuropathies are seen as well [74].

A cross-sectional study of newly diagnosed type 2 diabetics in Vietnam that studied the relationship between cardiovascular risk factors and the development of DPN found that smoking and poor control of HbA1C correlated with DPN, while body mass index, dyslipidemia, drinking, and hypertension did not have a direct relationship. Interestingly, the study found a statistically significant difference in levels of fasting GLP-1 patients who had developed DPN compared with patients who did not. The study found that there was, on average, an approximately 1.5 pmol/L decrease in GLP-1 levels among only the male patients who had DPN [49].

A review article by Bakbak et al. summarizes recently completed animal model trials that elucidated a relationship between GLP-1 agonism and the proliferation of new vasculature. The paper goes on to state that a limited number of trials have shown the GLP-1 agonism to be a mechanism to prevent endothelial dysfunction in type 2 diabetics with decreased peripheral circulation, a common factor leading to DPN [75].

Recent studies have shown a relationship between the prescription of GLP-1 agonists and an increase in cardiovascular health. The LEADER trial, which tested liraglutide, as well as the SUSTAIN-6 trial, testing semaglutide, compared these drugs to standard diabetic care via monitoring risk markers for cardiovascular disease [50,51]. Both studies showed that these medications lowered the risk of adverse cardiovascular events. While this is not a direct statement of these medications’ ability to reverse or prevent neuropathy in diabetics, it is possibly a sign that these medications may have a potential role in preventing new neuropathy via decreasing stress on the vasa nervorum. This may be because GLP-1 has been seen to decrease reactive oxygen species production and decrease vascular cell adhesion molecule-1 expression [76], in turn preventing endothelial inflammatory responses [77].

##### Nerve Fiber Repair

As previously mentioned, the pathogenesis of DPN is multifactorial, and all factors eventually coalesce as damage to nerve fibers. This damage in turn leads to the hyperexcitability of nociceptors, causing pain in patients with DPN. Several trials have studied the relationship between GLP-1 agonism and the possible respiration of the factors required in the process of repairing a damaged but not yet apoptotic nerve.

In one trial, rats were exposed to streptozotocin to induce diabetes. Diabetic rats were trialed on 1 nmol/kg/day doses of extendin-4, a GLP-1 agonist endogenous in gila monsters. The trial studied the perception threshold in the limbs of the treated rats as compared to rats not given extendin-4. Perception thresholds were quantified using a nerve stimulator set to a fixed frequency. The study found that over the course of 24 weeks, the control group not given extendin-4 gradually developed higher perception thresholds, while the experimental group maintained their initial values. After the trial concluded, the rats were sacrificed, and immunohistochemical staining of nerve fibers found that in extendin-4-treated rats, Schwann cell apoptosis was prevented and myelinated fiber size was maintained. Of note, the authors found that due to the near complete destruction of the pancreas via streptozotocin, the extendin-4 agonist was unable to modulate glucose control via increased insulin secretion. This would imply that the neuroprotection provided by extendin-4 in this trial is likely not due to the normal glucose-lowering effects of GLP-1 agonists, but rather direct agonism of the GLP-1 receptor on the nerve itself [78].

A similar trial was performed with liraglutide on rats that had streptozotocin-induced diabetes after 8 weeks of acclimation. Once again, the trial found that daily dosing of liraglutide for 8 weeks improved nerve health. The nerve conduction velocity was measured in the sciatic nerve of the rats throughout the trial. At the conclusion of the trial, the rats’ nerves showed signs of improved myelination under histological staining [79].

A 2015 pilot study randomized patients to exenatide or insulin glargine to evaluate the effect of exenatide on DPN symptoms in type 2 diabetics with DPN. The trial followed the progression of patient’s symptoms of DPN as well as their peripheral nerve conduction values and epidermal nerve fiber density. The trial concluded having found no significant differences in the GLP group as compared to the insulin patients in any of their measured categories [80].

A 2020 study randomized patients with poorly controlled T2D to receive exenatide and pioglitazone or insulin aspart and glargine. After following up with patients at the one-year mark, it was found that both groups showed corneal nerve regeneration compared to baseline. The study did not show any difference in DPN symptoms or metrics in either group compared to baseline [81].

In a 2021 study performed in Australia, patients receiving exenatide, a DPP-4 inhibitor, or a SGLT-2 had their motor nerve excitability tested as compared to healthy controls. The exenatide arm of the study was found to have normal nerve excitability as compared to DPP-4- or SGLT-2-treated patients. The researchers then wanted to further assess the effect of exenatide, and performed a prospective analysis with a smaller group of subjects. This second part of the study compared subjects before and 3 months after the start of treatment with exenatide. At the conclusion of the trial, the researchers found a statistically significant improvement in all three of their measures of nerve excitability as compared to baseline recordings in study subjects. Furthermore, the study found that there was no correlation between the percent improvement in HbA1c or blood pressure and the change in nerve excitability [52].

It should be noted that all of these trials have relatively small sample sizes, as the largest experimental group among all three trials was 90 patients. Furthermore, each trial recruited patients at different stages of progression in their DPN, indicating that the stage of pathogenesis potentially impacts the efficacy of the treatment.

Although these trials have mixed results, there are still no large-scale clinical trials of GLP-1 agonists in patients with DPN. Additionally, the mechanism by which GLP-1 agonists participate in nerve homeostasis has still not been fully elucidated. At this time, there is no definitive conclusion to be drawn as to if GLP-1 agonists are useful in the treatment of neuropathy, but the current research does offer promise in their usage looking forward.

#### 5.2.2. SGLT-2 Inhibitors

SGLT-2 inhibitors are another class of medications that have been theorized to have potential efficacy in the treatment of DPN. SGLT-2 inhibitors are used in the treatment of type 2 diabetes and act on the kidney to inhibit the reabsorption of glucose. Although there have been many clinical trials that have involved SGLT-2 inhibitors, the vast majority have not assessed outcomes related to DPN. As previously mentioned in this paper, meta-analyses have not found a correlation between the usage of SGLT-2 inhibitors and distal limb protectiveness. One analysis stated that GLP-1 agonists had a statistically significant reduction in adverse limb events as compared to SGLT-2 inhibitors [48]. This finding needs more supporting evidence, as another meta-analysis found no difference between SGLT-2 inhibitors and GLP-1 agonists when looking at limb amputation as a specific adverse limb outcome [73].

So far, no trials have looked at SGLT-2 inhibitors as a monotherapy for the treatment of DPN in diabetic patients. Looking to the future, there is currently one trial underway (NCT05162690) that looks to test the efficacy of dapagliflozin in DPN.

In animal models, a few studies have shown some promising preliminary data. in one trial, the SGLT-2 inhibitor ipragliflozin was tested on rats to determine the effects on DPN. The rats were diabetic prior to the start of testing, and non-diabetic rats were used as a control. The study found that the conduction velocity of peripheral leg nerves improved after SGLT-2 inhibitor treatment [82].

A study that used streptozotocin to induce diabetes in rats showed that SGLT2 inhibitor therapy may have a role in slowing down the pathogenesis of DPN via reversing risk factors such as oxidative stress, inflammation, and glucotoxicity [83]. Another trial, which used empagliflozin in diabetic rats, found that SGLT2 inhibitor treatment prevented the loss of skin nerve fibers and peripheral hypersensitivity [84].

Although these trials generally do provide insight into the possible efficacy of SGLT-2 inhibitors in limiting DPN, the lack of clinical trials means that conclusions cannot be drawn as to the efficacy of these medications in humans.

## 6. Clinical Trials

As previously established, the chronic nature, high prevalence, and severity of pain in DPN makes this clinical entity a burden to both patients and the healthcare system. Taking this into account, it is safe to say that the search for the improvement of symptoms and quality of life in these patients is continuous. A vast array of clinical trials that are currently recruiting or have recently been completed show promising approaches to treatment. From opioid agonists, anti-inflammatory and anti-oxidant treatment to fusion proteins and vitamin analogs, future clinical trials are covering extensive possibilities for pain alleviation.

It should be noted that most of the recently completed clinical trials focus on the use of newer and previously less explored medication groups, such as monoclonal antibodies and different gene-recombinant products. These modern advances in pharmacological therapy have shown great promise in the field of DPN pain, taking into account some of the currently published results.

One of the completed trials was in phase 3, while the rest were completed in phase 2. The trials followed different measurements for classifying pain relief and management, including a 50% reduction in average daily pain, change from baseline in different pain scores, or daily interference scores. Some of the trials found the medication to be more safe than effective, whereas others could not draw a concrete conclusion. This is in part due to the fact that, as previously described in the review, the pathophysiology of DPN pain is multifactorial, complex, and highly individual among patients, making it harder to find an all-encompassing treatment regime.

Summaries of currently recruiting and completed clinical trials concerning DPN pain are shown in Table 2 and Table 3, respectively.

## 7. Conclusions

In conclusion, we have summarized the current common pharmacological treatment options available for the treatment of diabetic peripheral neuropathy. We have also raised questions about the possible efficacy of treatments outside of the standard of care recommendations made by the ADA and AAN, such as capsaicin, GLP-1 agonists, and mirogabalin. Although GLP-1 agonists do not yet have enough trials to argue they are effective in treating diabetic peripheral neuropathy, they could be a promising intervention. Looking to the future, more research is needed to determine which patients respond best to each of these medications. Pharmacogenetic trials can help to understand why many patients must trial multiple medications before finding significant relief. Furthermore, it is unclear when in the pathogenesis of DPN a patient could be eligible for polypharmacy treatment plans, and future research can help to indicate the efficacy and interactions of multiple DPN medications together.

## Figures and Tables

**Table 1 pharmaceuticals-16-00801-t001:** Table of mentioned systematic reviews, meta-analyses and important studies.

Name/Reference	Type	Year	RCTs/Studies	Patients	Treatment/Intervention/Measurement	Outcomes
Lunn et al. [35]	SR	2014	18	6407	Duloxetine: 60, 120 mg/day	Primary: Short-term improvement in pain
						Secondary: Long-term improvement in pain, improvement in quality of life score, patient-reported pain, adverse effects during treatment
Yuan-Chun Ko et al. [36]	SR and MA	2021	3	290	Duloxetine: 20–80 mg/day Gabapentin: 300–1200 mg/day	Primary: VAS (Visual Analogue Scale) Secondary: Sleep Interference Score, Clinical Global Impression of Change, Patient Global Impression of Change, DN Symptom Score, DN Examination Score, Neuropathic Disability score
Chung-Sheng Wu et al. [37]	SR and MA	2023	7	2205	Duloxetine: 20–120 mg/day	Pain improvement, patient-reported health performance and quality of life
Andreas Limpas et al. [38]	SR	2021	83	/	Anticonvulsants, SNRIs, TCAs, opioids, topical treatment, cannabinoids, monoclonal antibodies, botulinum toxin, other	/
Floortje van Nooten et al. [39]	SR and MA	2017	24	5870	Capsaicin 8%	At least 30% pain reduction, at least 50% pain reduction, tolerability
Aaron Vinik et al. [40]	R, DB, Comparator-Controlled Study	2014	/	452	Mirogabalin: 5–30 mg/day	Primary: ADPS (Average Daily Pain Score) change from baseline Secondary: Characterizing dose response, incidence of responders, comparing effects of mirogabalin to pregabalin, assessing time to meaningful pain relief
Masayuki Baba et al. [41]	RA, DB, PC Study	2019	/	834	Mirogabalin: 15–30 mg/day	Efficacy, safety, and tolerability
Titas Buksnys et al. [42]	SR and MA	2020	43	/	Lidocaine medicated plaster 700 mg	Efficacy, adverse effects
Moisset et al. [43]	SR	2020	131	/	TCAs, SNRIs, antiepileptics, opioids, topical agents, cannabinoids, ketamine, other	Comprehensive assessment of all therapies for neuropathic pain treatment
Farag Hussein et al. [44]	SR and MA	2022	36	11,930	Duloxetine: 60 and 120 mg/day Pregabalin: 150–600 mg/day Milnacipran: 100 and 200 mg/day Amitriptyline	Comparative effectiveness and acceptability of medication for pain, sleep, depression, fatigue, and quality of life
Nanna Finnerup et al. [45]	SR and MA	2015	229	/	TCAs, SNRIs, antiepileptics, opioids, oromucosal cannabinoids, topical lidocaine, capsaicin patches, other	Individual and combined number needed to treat and number needed to harm values
Solomon Tesfaye et al. [46]	R, DB, Multicenter, Crossover Trial	2022	/	130		Primary: Difference in 7-day average NRS (Numerical Rating Scale) daily pain Secondary: HADS (Hospital Anxiety and Depression Scale), proportion of patients achieving 30% and 50% pain reduction from baseline, ISI (Insomnia Severity Index), NPSI (Neuropathic Pain Symptom Inventory), other
Zin Zin Htike et al. [47]	SR and Mixed-Treatment Comparison Analysis	2017	34	14,464	Glucagon-like peptide-1 receptor agonist (GLP-1ARs): albiglutide, dulaglutide, exenatide, liraglutide, others	Glycemic control, body weight, blood pressure and lipid profile, gastrointestinal and other side effects
Donna Shu-Han Lin et al. [48]	Retrospective Cohort	2022	/	101,440	Glucagon-like peptide-1 receptor agonist (GLP-1ARs); Sodium-glucose cotransporter 2 inhibitors (SGLT2is)	Primary: Major adverse limb events (MALE) Secondary: Major adverse cardiac events (MACE), death from any cause, hospitalization due to heart failure
Tuan Dinh Le et al. [49]	Cross-sectional	2022	/	473	GLP-1 serum levels	Prevalence of DPN and its risk factors, relation between DPN and fasting GLP-1 levels
Steven Marso et al. [50]	R, DB Trial	2016	/	9340	Liraglutide 1.8 mg/day	Fist occurrence of death from cardiovascular causes, non-fatal MI, or non-fatal stroke, microvascular outcomes (renal and retinal), neoplasms, pancreatitis
Steven Marso et al. [51]	R, DB Trial	2016	/	3297	Semaglutide 0.5 or 1.0 mg/week	Fist occurrence of death from cardiovascular causes, non-fatal MI, or non-fatal stroke, microvascular outcomes (renal and retinal)
Tushar Issar et al. [52]	Cross-sectional	2021	/	90	GLP-1RA, DPP-4i, SGLT-2i	Improvement in nerve excitability

SA—systematic analysis; MA—meta-analysis; R—randomized; DB—double-blind; PC—placebo-controlled.

**Table 2 pharmaceuticals-16-00801-t002:** Table of completed trials of importance for DPN.

Sponsor	NCT Number	Study Description	Phase	Intervention	Total #	Per Arm	Description of Results	Results
Helixmith Co.	NCT02427464	R, DB, PC, Multicenter Study	3	Engensis (VM202) (plasmid DNA encoding hepatocyte growth factor (HGF))	500	336 (VM202 0.5 mL inj.)	Participants with at least 50% reduction in average 24 h pain score from baseline on day 90	69 (20.5%)
								28 (17.1%)
				Placebo		164 (Placebo)	Participants with at least 50% reduction in average 24 h pain score from baseline on day 180	113 (33.6%)
								42 (25.6%)
Pfizer	NCT01087203	R, DB, PC, Multicenter Study	2	Tanezumab (monoclonal antibody against nerve growth factor)	73	38 (tanezumab 20 mg inj.)	Change from baseline in average diabetic peripheral neuropathy (DPN) pain score in week 16 (shown as mean (SD))	−1.04 (1.92)
				Placebo		35 (Placebo)		−2.10 (3.14)
Eli Lilly and Co.	NCT04476108	R, PC, Parallel Assignment Trial	2	LY3016859 (anti-TGFA recombinant antibody)	125	84 (LY3016859 750 mg)	Change from baseline in average pain intensity as measured using the NRS (shown as mean (95% CI))	−1.98 (−2.42 to 1.55)
								−1.56 (−2.17 to −0.96)
				Placebo		41(Placebo)	Change from baseline in the Brief Pain Inventory–Short Form (BPI-SF) total interference score (shown as mean (95% CI))	−2.11 (−2.55 to −1.65)
								−1.74 (−2.35 to −1.12)
Daiichi Sankyo and Co.	NCT01496365	R, DB, PC, Parallel Assignment Study	2	DS-5565 (Mirogabalin)	216	112 (placebo)	Mean change from baseline to week 5 in Average Daily Pain Score (ADPS) following treatment with DS-5565 compared to pregabalin and placebo (shown as mean (SD))	−1.86 (2.18)
						56 (pregabalin 150 mg BID)		−1.79 (2.27)
				Pregabalin		57 (DS-5565 5 mg QD)		−2.04 (2.22)
						57 (DS-5565 10 mg QD)		−2.32 (2.17)
						57 (DS-5565 15 mg QD)		−2.66 (2.37)
				Placebo		56 (DS-5565 10 mg BID)		2.64 (2.45)
						57 (DS-5565 15 mg BID)		−2.79 (2.43)

R—randomized; DB—double-blind; PC—placebo-controlled.

**Table 3 pharmaceuticals-16-00801-t003:** Table of recruiting trials of importance for DPN pain.

Sponsor/PI	NCT Number	Study Description	# of Participants	Phase	Intervention	Drug Group
Vrooman et al.	NCT04678895	R, DB, PC, Crossover Trial	35	2	Naltrexone Placebo	Opioid antagonist
AstraZeneca	NCT03755934	R, DB, PC, Dose–Response Study	111	2	MEDI7352 Placebo	Fusion protein binding nerve growth factor (NGF) to tumor necrosis factor receptor 2 (TNFR2)
Rathmell et al.	NCT05480228	Prospective, Parallel Group, Multicenter, R, DB, PC	122	2	NRD135SE.1 Placebo	Non-opioid molecule with unknown target
Basit et al.	NCT05080530	Non-R, No Masking (Open-Label)	216	N/A	Cholecalciferol	Vitamin D analog
Vertex Pharmaceuticals	NCT05660538	R, DB, Active-Controlled, Dose-Ranging, Parallel Design Study	175	2	VX-548 Pregabalin Placebo	Selective NaV1.8 sodium channel inhibitor GABA analog
Mittendorfer et al.	NCT05145452	R, DB, Controlled Trial	60	N/A	Fish oil-derived n-3 polyunsaturated fatty acids	Lipid-regulating agent
Eli Lilly and Co.	NCT05620576	R, PC Master Protocol	125	2	LY3857210 Placebo	P2XY inhibitor
Elsharab et al.	NCT05369793	R, No Masking (Open-Label)	60	3	Alpha-lipoic acid Roflumilast	Antioxidant Phosphodiesterase inhibitor
Zhao et al.	NCT05507697	R, Open-Label, Single-Center Trial	42	2	HUC-MSCs Lipoic acid	Stem cells
Rastogi et al.	NCT05162690	R, PC, Double Masking Trial	40	3	Dapagliflozin Placebo	Sodium-glucose cotransporter 2 (SGLT2) inhibitor
Emara et al.	NCT04766450	R, Open-Label, Controlled Trial	30	4	Acetyl-cysteine	Anti-oxidant
Ameo et al.	NCT05247034	R, DB, Controlled Trial	5	N/A	Cocoa supplement	Anti-inflammatory and anti-oxidant
Wang et al.	NCT04457531	R, Open-Label, Controlled Study	60	1	LiuWeiLuoBi granule	Anti-inflammatory

R—randomized; DB—double-blind; PC—placebo-controlled.

## Data Availability

Not applicable.

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
