# Peer review of "A Review of Recent Pharmacological Advances in the Management of Diabetes-Associated Peripheral Neuropathy"

_pharmaceuticals, 2023, doi:10.3390/ph16060801_

Round 1
Reviewer 1 Report
This is a well written and up to date review on DPN.
1. Name the OPTION-DM study, rather than just a recent randomized study in 2022. Also highlight the overall poor response (40-50% pain relief) for all three therapies (pregabalin, duloxetine and amitriptyline). This urges the need for newer therapies.
2. State that TCA’s and Oxcarbazepine have Na channel blockade action.
3. Tirzepatide is a dual GLP-1/GIP agonist, need to differentiate it from the established GLP-1 agonists. Also consider the new triple agonists being evaluated.
4. What about the effect of SGLT2-i on neuropathy.
5. Consider the two clinical trials which have evaluated the effect of GLP-1 therapy on underlying nerve damage in DPN (Jaiswal et al J Diab Comp 2015; 29: 1287-1294; Ponirakis et al. BMJ Open DRC 2020; 8: e001420).
Overall well written.
English could be made more succinct.
Author Response
- Reviewer’s comment: “Name the OPTION-DM study, rather than just a recent randomized study in 2022. Also highlight the overall poor response (40-50% pain relief) for all three therapies (pregabalin, duloxetine and amitriptyline). This urges the need for newer therapies.”
Response: Thank you for your comment. I agree that we could have clarified the response to therapy in the different trial groups. We have edited the paper to properly state the name of the trial as well as made a note of the inadequate response to monotherapy and combination therapy for a significant number of patients. We appreciate the suggestion and agree that it makes a stronger statement about the need for newer therapies.
- Reviewer’s comment: “State that TCA’s and Oxcarbazepine have Na channel blockade action.”
Response: We have edited the relevant paragraph to state the important fact that many sodium channel blockers are also classified under the heading of TCAs or anticonvulsants, such as oxcarbazepine. Thank you for this comment.
- Reviewer’s comment: “Tirzepatide is a dual GLP-1/GIP agonist, need to differentiate it from the established GLP-1 agonists. Also consider the new triple agonists being evaluated.”
Response: Thank you for this comment. This is an important note that does differentiate tirzepatide in its mechanism of action and pharmacodynamic properties on the body. We have made a subsequent edit to account for this fact. We also mentioned retatrutide, a triple receptor agonist. Because of the limited amount of information on retatrutide currently, we have refrained from making any assumptions on its possible difference in action in the treatment of DPN.
- Reviewer’s comment: “What about the effect of SGLT2-i on neuropathy.”
Response: SGLT2 inhibitors were initially mentioned only in brief as there is a paucity of clinical trials on their effect as related to DPN. We do acknowledge that they are important medications, and a trial is currently ongoing. We apologize for this oversight and have added a brief section that goes over a few animal model trials that have shown interesting results. Thank you for this comment.
- Reviewer’s comment: “Consider the two clinical trials which have evaluated the effect of GLP-1 therapy on underlying nerve damage in DPN (Jaiswal et al J Diab Comp 2015; 29: 1287-1294; Ponirakis et al. BMJ Open DRC 2020; 8: e001420).”
Response: Thank you for bringing these articles to our attention, we have edited the paper to include their information. As these papers find no clinical difference between their treatment arms, we feel that their inclusion raises more questions about the type of DPN patient that GLP-1 agonist treatment may be effective in. We have also, therefore, added a minor paragraph that alerts readers to the fact that all three trials are relatively small and have recruited patients with slightly different clinical presentations. Thank you.
Reviewer 2 Report
The authors reviewed recent studies regarding the pharmacological interventions toward diabetic peripheral neuropathy (DPN). The manuscript is basically well-written and informative, but the following issues need to be addressed.
1. Although the title suggests a review for wide ranges of medicines against DPN, the authors focused merely on the remedies for painful diabetic neuropathy (SNRIs, gabapentinoids, sodium channel blockers, and TCAs) and GLP-1 mimetics. In addition, it is better to categorize the remedies into symptomatic and pathogenesis-based treatments.
2. The authors stated that tetrodotoxin (TTX) non-selectively block all sodium channels regardless of their state (Line 202), but the channels are classified into TTX-sensitive and -resistant ones. In addition, the clinical trials of selective Nav1.7 channel blockers should be mentioned.
3. Although autonomic neuropathy is a major symptom of DPN, the author totally ignored it.
4. Because numerous abbreviations are used in the text, it is recommended to make a list of them. DN (Lines 30-33) stands for diabetic neuropathy? Are there any differences between DPN and DN? What does GRP (Line 249) stand for?
Author Response
- Reviewer’s comment: “Although the title suggests a review for wide ranges of medicines against DPN, the authors focused merely on the remedies for painful diabetic neuropathy (SNRIs, gabapentinoids, sodium channel blockers, and TCAs) and GLP-1 mimetics. In addition, it is better to categorize the remedies into symptomatic and pathogenesis-based treatments.”
Response: We thank you and the reviewer for comments and criticisms which will definitely improve the manuscript. We have considered each comment carefully and have revised our manuscript to address all raised points. We understand how the title of the paper might have been misleading, alluding to the fact that we would be covering all the possible treatment options for diabetic neuropathy. However, the paper is solely focused on pain and pain management, therefore the main entity we covered was painful diabetic neuropathy. We felt that doing a review of all treatment modalities for different forms of diabetic neuropathy would have been too extensive and that the paper would lose its integrity and focus.
Thank you for the suggestion regarding the categorization of treatment options into two main groups. Changes have been made to make it more visible and prominent in the paper.
- Reviewer’s comment: “The authors stated that tetrodotoxin (TTX) non-selectively block all sodium channels regardless of their state (Line 202), but the channels are classified into TTX-sensitive and -resistant ones. In addition, the clinical trials of selective Nav1.7 channel blockers should be mentioned.”
Response: Thank you for this comment, as this is a missed mark on our part. We have corrected the text, added more precise information regarding tetrodotoxin and included several references. We have also included new information about Nav1.7 channel blockers, followed by some of the newer and more relevant studies.
- Reviewer’s comment: “Although autonomic neuropathy is a major symptom of DPN, the author totally ignored it.”
Response: Thank you for your suggestion, we agree with you completely that autonomic neuropathy is a big part of DPN presentation. However, the aim of this paper was mostly on diabetic sensorimotor neuropathy, its painful presentation, and the management of said pain. With that, we made the decision to not include other forms of diabetic neuropathies, that we briefly mentioned in our introduction, such as mononeuropathies, polyneuropathies and autonomic neuropathies. Reviewing treatment options for all diabetic complications would have been an extensive topic and something that we concluded would have not been in the scope of this paper. Dedicating a specific part of the paper to autonomic neuropathy would question the importance of all other forms of diabetic neuropathies and would call for explanations of those as well. We do hope you understand our thinking process here and that you will be satisfied with our response.
4 Reviewer’s comment: “Because numerous abbreviations are used in the text, it is recommended to make a list of them. DN (Lines 30-33) stands for diabetic neuropathy? Are there any differences between DPN and DN? What does GRP (Line 249) stand for?”
Response: Once again, we are very grateful for your insight, since this is something that will better the paper entirely. We have added an Appendix A at the end of the paper listing all the abbreviations that have been used in the paper.
To answer your question, we used DN (diabetic neuropathy) in a greater sense, encompassing DPN (diabetic peripheral neuropathy) in it. DPN is preferably used throughout the paper because it is the peripheral damage that we covered in the paper and suggested treatments for. That is why DN was briefly mentioned just in the introduction of the paper.
Round 2
Reviewer 2 Report
The author revised the manuscript in an appropriate manner, and it is worth reading.